# Functional Microfiber Nonwoven Fabric with Sialic Acid-Immobilized Polymer Brush for Capturing Lectin in Aerosol

**DOI:** 10.3390/polym14040663

**Published:** 2022-02-09

**Authors:** Yung-Yoon Kim, Kanta Sagara, Kazuya Uezu

**Affiliations:** 1Graduate School of Environmental Engineering, The University of Kitakyushu, 1-1 Hibikino, Kitakyushu 808-0135, Japan; z8dab002@eng.kitakyu-u.ac.jp (Y.-Y.K.); c1mab007@eng.kitakyu-u.ac.jp (K.S.); 2Faculty of Environmental Engineering, The University of Kitakyushu, 1-1 Hibikino, Kitakyushu 808-0135, Japan

**Keywords:** radiation-induced graft polymerization, influenza virus, lectin, WGA, iminodiacetate, microfiber nonwoven fabric, *N*-acetylneuraminic acid

## Abstract

The influenza virus has been known as a representative infectious virus that harms human health from the past to the present day. We have promoted the development of a novel adsorbent capable of adsorbing influenza viruses in the form of aerosols in the air. In this study, to develop a material to adsorb the influenza virus, a functional group was introduced into a microfiber nonwoven fabric (MNWF) manufactured through radiation-induced graft polymerization (RIGP), and sialic acid was immobilized to mimic the sugar chain cluster effect. The functional group was used by coupling disodium iminodiacetate monohydrate (IDA) and 1-(3-dimethylaminopropyl)-3-ethylcarbodiimide hydrochloride (EDC), and *N*-acetylneuraminic acid (NANA) was selected for sialic acid. IDA-EDC was introduced into GMA MNWF with an average molar conversion of 47%. For NANA MNWF with a degree of grafting (*dg*) of 87% introduced with sialic acid, 118.2 of 200 µg of aerosolized lectin was adsorbed, confirming that the maximum adsorption amount was 59.1%. In NANA MNWF of 100% or more *dg*, a tendency to decrease the amount of lectin adsorption was observed compared to NANA MNWF of 80–100% *dg*.

## 1. Introduction

It is no exaggeration to say that viruses accompany the history of humanity. In particular, even now, infectious viruses are a severe problem that affects the economy of the country and the lives of the people. There are countless infectious viruses, but the most representative is the influenza virus, which has been recognized as a big problem from the past to the present. Influenza infection, which is a representative contagious virus, usually starts in winter, in the eastern or southern hemisphere with high humidity, and then tends to spread to North America and Europe. In particular, when mutations occur, infectious diseases spread more [1]. In the case of influenza, there is an incubation period of about three days, and during this period, the virus carrier spreads the virus because there are no symptoms. Especially before the 1950s, influenza pandemics killed up to millions to tens of millions of people before influenza was found to be caused by a virus and a vaccine was developed [1,2]. In the modern era of active vaccine development, dying from influenza is lower than before. However, deaths are still occurring. Infectious viruses are still very deadly and dangerous because they evolve with mutants. Vaccine research is active, however influenza viruses are constantly mutating, and the most effective treatment is vaccination and isolation from the source of infection.

In general, the route of infection of infectious viruses can be divided into infection by droplet and infection by contact [3,4]. Infection by contact occurs when the mucous membrane is touched with the part of the body that has come into contact with the virus, and the best way to prevent this is to keep the body clean. However, since infection by droplets is invisible, it is difficult for both carriers and those around them to notice the risk of infection, and it is even more dangerous. Both droplet and contact infections occur when the virus binds to cells in the human body, and hemagglutinin (HA) [3,5,6,7] and neuraminidase (NA) are known to be involved in the primary infection routes of the infectious virus. HA is one of the antigenic projections, a type of glycoprotein that causes red blood cells to aggregate, and NA is one of the enzymes that are antigenic determinants found on the surface of influenza viruses in various organisms. Among them, HA is a direct factor in the infection process as it causes the antigenic projections it possesses to directly attach to the cells of animals and plants. Influenza viruses (80–120 nm) [8] are known to cause infection by binding to sialic acid [6,7,9], a type of sugar on the cell surface. Sialic acid present on the surface of animal and plant cells generally exists in the form of a sugar chain. Sugar chains refer to monosaccharides that combine in various formats to form a chain shape. Monosaccharides that make up sugar chains include glucose, mannose, galactose, fucose, xylose, *N-*acetylglucosamine, and sialic acid, and these sugar chains interact with a single protein. In general, the interaction between a single sugar chain and a single protein is weak because weak forces such as hydrogen bonding and hydrophobic interactions mainly occur. Compensating for these shortcomings is the sugar cluster effect or sugar chain polyvalent effect [10]. In other words, proteins or glycolipids do not spread into cells but concentrate in some areas as needed to form sugar clusters, increasing the binding force. This sugar cluster effect has become the primary design guideline for developing antibodies, antivirals, and toxin inhibitors in immunology.

We decided to mimic the sugar chain cluster effect as way of developing material for adsorbing the influenza virus, and MNWF prepared by radiation-induced graft polymerization (RIGP) was selected as the substrate. RIGP introduces various functions using radicals generated when a base substrate is irradiated with radiation, plasma, light, and chemicals. The main base polymers used are polyethylene, polypropylene, polytetrafluoroethylene, nylon 6, etc. [11], and the various types of base polymers applied here include porous hollow fiber membranes, non-woven fabrics, films, and nanotubes [12,13,14,15]. In particular, substrates made through RIGP have the advantage of controlling the length or density of polymer brushes created by controlling the amount of radiation or monomer concentration [11]. Polymer brush is one of the characteristics of polymer chains generated by RIGP and refers to polymer chains extending from the pore surface of the base substrate to the interior of the substrate [11]. We expected that if sialic acid could be introduced into the polymer brushes of the substrate prepared by this radiation graft polymerization method, it would be possible to mimic the effect of sugar chain clusters by controlling the density of the polymer brushes. In general, the average length of polymer brushes extending to the substrate is known to be 35 nm [12], which is a sufficient size to bind HA (13.5 nm) [5,6] present on the surface of the influenza virus. Glycidyl methacrylate (GMA) was selected as the monomer used for RIGP, and in this case, the polymer contains an epoxy group and has hydrophilicity [12,13,14]. For the polymer base material, we selected microfiber nonwoven fabric (MNWF) made of polypropylene. In general, the advantage of nonwoven fabrics is that they have large pores compared to other substrates, so they are not easily clogged and are durable [14]. The disadvantage is that the functional group bonded to the nonwoven fabric has a smaller surface area than other substrates in contact with the adsorption target, so it may be disadvantageous for adsorption. However, MNWF used in this experiment was selected as a material for the experiment because it can overcome the disadvantage of a narrow surface area by weaving with microfibers while maintaining the advantages of the existing nonwoven fabric.

This experiment aims to target the selective adsorption of hemagglutinin on the surface of infectious viruses by mimicking the effect of sugar chain clusters through the RIGP method. However, we did not use it directly because of the risk of infection when handling the influenza virus as an early step in influenza virus adsorption experiments. Instead, an experiment was performed by selecting a lectin with a similar binding reaction through a specific recognition with sialic acid. Therefore, the efficiency of lectin binding was checked by introducing sialic acid into MNWF with various polymer brush concentrations. Figure 1 shows a projected image of the MNWF attempting influenza virus adsorption with sialic acid introduced into a polymer brush.

## 2. Materials and Methods

### 2.1. Materials

The following materials were used: disodium iminodiacetate monohydrate (98%, WAKO Chemical Co., Ltd., Odawara, Japan), microfiber nonwoven fabric (polypropylene, *dg* 38~183%, ENEOS Corporation, Tokyo, Japan), 1-(3-dimethylaminopropyl)-3-ethylcarbodiimide hydrochloride (98%, Tokyo chemical industry Co., Ltd., Tokyo, Japan), *N*-acetylneuraminic acid (Nacalai tesque, INC, Kyoto, Japan), WGA-Biotin (J220, J-chemical, INC, Tokyo, Japan), sodium dodecyl sulfate (95%, WAKO Chemical Co., Ltd., Odawara, Japan), BCA Protein Assay Kit (T9300A, Takara, Kusatsu, Japan), Synergy HT Multimode Microplate Readers (Biotek, Winooski, VT, USA), Spectrum Two FT-IR Spectrometer (Perkin Elmer, Waltham, MA, USA), wheat germ agglutinin, Triticum vulgaris, FITC Conjugate (FL-1021, Vector Laboratories, Burlingame, CA, USA), Olympus IX81 fluorescence microscope (Olympus Optical Co. Ltd., Tokyo, Japan), and FE-SEM-EDS (JSM-7800f, Jeol, Tokyo, Japan).

### 2.2. Methods

#### 2.2.1. Selecting Target Lectin

Since we are in the initial stage of the virus adsorption experiment, we had to select an appropriate level of the target. Using the influenza virus directly in adsorption experiments is challenging, and other methods had to be devised. Therefore, as an alternative to the influenza virus used in adsorption experiments, we decided to target a lectin that can explicitly react with sialic acids, such as the HA of the influenza virus. *N*-acetylneuraminic acid (NANA), which is used in this experiment, is one of the sialic acids and is known to respond with H1-type hemagglutinin of the influenza virus by specific recognition [16,17,18,19,20,21,22]. Similar to the specific recognition reactions of NANA with HA, concanavalin A (ConA) [16,21] and wheat germ agglutinin (WGA) [16,21,22] are also capable of specific recognition reactions with NANA. Therefore, rather than directly adsorbing the influenza virus in the design stage of the adsorption experiment, we tried to check whether the lectin, which is a non-hazardous material, can be adsorbed on the newly developed adsorbent. Con A and WGA were candidates for the lectin to be used in the experiment. However, in our preliminary experiments, WGA had better adsorption performance than Con A, so the adsorption target lectin was finally determined as WGA. Although the crystal structures of the influenza virus and WGA are different, both can bind by NANA-specific recognition and are considered helpful as alternatives to determining whether the influenza virus is adsorbed or not without infection risk during the experiment. WGA lectins are by-products of wheat processing, whose structure has been described since the 1970s [23,24]. WGA lectins are composed of polypeptide chain homodimers [25] and are in the form of biaxial symmetry [26]. Polypeptides A, B, and C present in WGA are known to bind to *N*-acetylglucosamine (GlcNAc) residues [23,27]. Figure 1 shows the crystal structure of the influenza virus [28], which caused the worst pandemic in history in 1918, and the crystal structure of the WGA lectin.

#### 2.2.2. Mimic of Sugar Cluster Effect of Functional MNWF

It has already been demonstrated in several studies whether an artificial enzyme can function as a catalyst by immobilizing it on a polymer brush [12,29,30]. Thus, it was expected that the introduction of sialic acid by selecting an appropriate functional group could be efficiently used for lectin binding. The selection of functional groups was considered from various perspectives, and the functional group used in this experiment was determined to be disodium iminodiacetate monohydrate (IDA). The IDA group is a functional group [31,32,33,34,35] widely used in protein adsorption, enzyme purification and fixation, and metal ion adsorption. In particular, IDA is judged to be more efficient when introducing sialic acid because there are two places where sialic acid can be introduced. However, in this experiment, to increase the efficiency of introducing sialic acid, it was finally decided to use the IDA-EDC functional group made through coupling with 1-(3-dimethylaminopropyl)-3-ethylcarbodiimide hydrochloride (EDC) instead of using IDA alone. The coupling process of IDA-EDC is shown in Figure 2. In general, EDC is a substance used to covalently bond carboxylate (–COOH) or phosphate (–PO_4_) sulfide with an amine group (–NH_2_). EDC has the disadvantage of being self-polymerized in some target biomolecules (e.g., peptides) containing both carboxylates and amine groups, however, it is water-soluble, so it is easy to handle and efficiently reacts with amine compounds [36].

IDA was used at a concentration of 0.125 mol, and since IDA has two sites that can react with sialic acid, EDC was used at a concentration of 0.25 mol, twice the concentration of IDA, to prepare an IDA-EDC complex. In the case of the IDA-EDC complex, a coupling reaction was performed at 25 °C for 72 h before being introduced into MNWF and used. Similarly, *N*-acetylneuraminic acid (NANA, sialic acid) was also introduced at a concentration of 0.25 mol, twice the concentration of IDA.

ENEOS Corporation provided the GMA MNWF used in this experiment, and the concentration of GMA used was 3–8.5%, and the irradiation dose was 10~20 kGy. The degree of grafting (*dg*) used in the experiment was 38–173%. The equations for the *dg* and the amount of GMA introduced are:(1)$$\mathrm{Degree}\;\mathrm{of}\;\mathrm{graft}\;\left(dg\right)=100\left(W_{1}-W_{0}\right)/W_{0}\;\left[\%\right]\;$$
(2)$$\mathrm{Introduced}\;\mathrm{GMA}=\left[1000\left(W_{1}-W_{0}\;\right)⁄\left(142.2\right)\right]⁄W_{0}\left[\mathrm{mmol}⁄g\right]$$

Here, W_0_ is the mass of the polymer that has not been treated, and W_1_ is the mass of the polymer after reacting with the GMA monomer [11].

The amount of IDA-EDC introduced into GMA MNWF was calculated through the weight change of each sample and the molar conversion rate [11]. The equation of the molar conversion rate is:(3)$$\mathrm{Molar}\;\mathrm{conversion}\;\mathrm{rate}=100\left[\left(W_{2}-W_{1}\right)/514.5]/[\left({\;W}_{1}-W_{0}\right)/142.2\right]\;\left[\%\right]$$
where W_0_ is the mass of the polymer that has not been treated, W1 is the mass of the polymer after reacting with the GMA monomer, and W2 is the mass of the polymer after the introduction of the functional group by the addition of an IDA-EDC to the epoxy group. The value of 142.2 is the molecular weight of GMA, and the value of 514.5 is the IDA molecular weight plus two EDC molecular weights. At each stage of the production, the functional nonwoven fabrics underwent sufficient washing and drying time to remove residual substances after the reaction. The MNWF into which each functional group was introduced was measured via FT-IR (range 4000–450 cm^−1^). The path through which IDA-EDC functional groups and NANA are introduced into the GMA MNWF is shown in Figure 3, and Table 1 summarizes the introduction conditions.

#### 2.2.3. Aerosol Spray Type Lectin Adsorption

Several methods have been devised to adsorb lectin onto NANA MNWF. Experiments for the development of adsorbents have traditionally used batch methods. When using the batch method, it was thought that the maximum lectin adsorption characteristics of NANA MNWF could be confirmed. However, since influenza viruses are generally infected in droplets, it was judged that the experimental method through the batch method was not appropriate. Therefore, we decided to aerosolize lectin and try adsorption to NANA MNWF. A lectin adsorption experiment was prepared similarly to studies in [37,38] that adsorbed aerosols onto nonwoven samples. Figure 4 shows the form of lectin aerosolization and adsorption to NANA MNWF used in the experiment.

For the aerosol adsorption experiment, a WGA lectin solution prepared at a 2000 µg/mL concentration was prepared, and 0.1 mL of the WGA solution was put into the nebulizer. The nebulizer was operated using a pump, and the aerosolized WGA solution was permeated through the NANA MNWF for about 1 min. After the adsorption process finished, NANA MNWF was washed with ionized water to remove lectins that may remain on the surface of NANA MNWF without being adsorbed.

For the aerosol adsorption experiment, a WGA lectin solution was prepared at a 2000 µg/mL concentration, and 0.1 mL of the WGA solution was put into the nebulizer. Therefore, 0.1 mL of WGA lectin solution was considered to contain up to 200 µg of WGA. The nebulizer was operated using a pump, and the aerosolized WGA solution was permeated through the NANA MNWF for about 1 min. After the adsorption process finished, NANA MNWF was washed with ionized water to remove lectins that may remain on the surface of NANA MNWF without being adsorbed. Lectins adsorbed to NANA MNWF were separated by 0.2 wt.% sodium dodecyl sulfate (SDS) solution [39,40,41,42]. Lectins isolated by SDS were treated with the bicinchoninic acid (BCA) method [42,43,44,45] and then measured using a spectrophotometer for quantitative measurement.

## 3. Results and Discussion

### 3.1. Introduction of Functional Group, IDA-EDC and NANA, into GMA MNWF

IDA-EDC was introduced into GMA MNWF according to the *dg*, and the characteristics of IDA-EDC introduction according to the *dg* were confirmed. These results are shown in Figure 5. The amount of IDA-EDC introduced into GMA MNWF was confirmed using Equations (2) and (3). In the case of IDA-EDC MNWF, it was confirmed that the higher the grafting degree, the more IDA-EDC was introduced. In this experiment, the highest density of IDA-EDC introduced in GMA MNWF was about 8.4 mmol/g at 173% *dg*. This seems to be because as the dg increases, the GMA content increases so that IDA-EDC reacts with more GMA. In MNWF supplemented with IDA-EDC, the surface state of MNWF was significantly changed as the *dg* increased. In the case of IDA-EDC MNWF with a *dg* of 100% or less, the edges of the MNWF were slightly curled, and in the case of IDA-MNWF with a *dg* of 100% or more, the MNWF was severely bent. It seems that the durability of MNWF cannot be sustained when IDA-EDC is introduced in too large a quantity. When IDA-EDC was introduced into the GMA MNWF, the weight change increased from 40% for low *dg* MNWF to 150% for high *dg* MNWF. From all results, the molar conversion rate of IDA-EDC MNWF from MNWF grafted with GMA was confirmed to be about 47% on average.

NANA was prepared as a solution with the same 0.25 M concentration as EDC and was obtained by reacting at a temperature of 298 K for 24 h so as not to induce a change in properties while NANA was introduced into IDA-EDC MNWF. After NANA was introduced, the weight of IDA-EDC MNWF increased by about 12~27%, and no change in appearance was observed.

The spectrum of FT-IR was measured to confirm the introduction of a functional group in MNWF for each step. The spectral results of FT-IR are shown in Figure 6, and the interpretation of each peak is shown in Table 2. For GMA MNWF, peaks in epoxy and ester groups were found at 747, 841, 972, 1053, 1166, and 1238 cm^−1^, and peaks of C=O were found at 1726 cm^−1^ [46,47,48,49,50,51,52,53]. In the case of IDA-EDC MNWF, the peaks of epoxy and ester groups were reduced compared to GMA MNWF, and it was confirmed that new peaks of N–H were generated at peaks of 1554–1564 [48,54] and 1659–1662 cm^−1^ [46,51,54]. In addition, new peaks of the hydroxyl group (–OH) and N–H were generated at 3250–3400 cm^−1^ [47,48,49,50,51,52,53,55], confirming the introduction of IDA-EDC. In the case of NANA MNWF, the peak indicating C=O and the carboxyl group (COOH) was observed at 1726 cm^−1^ [47,48,49,50,51,52,53], significantly increased, and the hydroxyl group of 3250–3400 cm^−1^ [47,48,49,50,51,52,53,55] was also increased when compared with the peak of IDA-EDC MNWF. The introduction of NANA was confirmed by increasing the peak size of the carboxyl group and the hydroxyl group.

In addition, the SEM image was confirmed to observe the change of the surface of the MNWF into which each functional group was introduced. Each sample was taken at 2200 magnification. The SEM image in which each functional group was introduced is shown in Figure 7. After GMA was initially introduced as a functional group, no significant difference was observed on the surface of MNWF while each IDA-EDC or NANA was introduced.

### 3.2. Lectin Binding Properties

The adsorption of the lectin was checked by using the fluorescently labeled lectin on the functional nonwoven fabric of each step. To confirm the adsorption of the lectin, the lectin fluorescently labeled with Fluorescein isothiocyanate [56,57,58] was adsorbed and eluted in the form of an aerosol. The results of fluorescence microscopy are shown in Figure 8. Fluorescence images were identified at a 4 magnification using an Olympus IX81 model.

As a result of confirming GMA MNWF through a fluorescence microscope, it was confirmed that almost no lectin was adsorbed. The fluorescence image could confirm a slight luminescence, but similar results were obtained even after lectin elution with 0.2 wt.% SDS solution. Therefore, it is thought that non-selective adsorption of lectin by GMA MNWF occurred. The IDA-EDC MNWF confirmed that the lectin was adsorbed to some extent through a slight light emission. Similarly, it was confirmed that a small amount of fluorescent material remained after the elution of the lectin. NANA MNWF showed the most robust luminescence image, ensuring the most effective lectin adsorption. In the case of NANA MNWF, it was confirmed that a small amount of fluorescent material also remained even after lectin was extracted with SDS solution. Through fluorescence microscopy analysis, NANA MNWF showed the most robust luminescence image. Still, IDA-EDC MNWF was also able to adsorb lectins, so the amount of lectin adsorption of MNWF introduced with each functional group was confirmed.

Figure 9 shows the lectin adsorption amount of GMA, IDA-EDC, and NANA MNWF at an average *dg* of 90%. As confirmed by the fluorescence image, it was confirmed that a minimal amount of non-selective lectin adsorption occurred in GMA MNWF. In the case of IDA-EDC, lectin adsorption was possible to some extent. However, it was shown at half the level of NANA MNWF, confirming that the sialic acid was properly introduced into NANA MNWF functions.

As NANA MNWF showed utility for lectin adsorption, the lectin adsorption amount change of NANA MNWF according to the *dg* was compared. The difference in the amount of lectin adsorption according to the *dg* is shown in Figure 10.

The maximum lectin adsorption amount was 59.1% in *dg* 87% NANA MNWF. This is the result of adsorption of 118.2 out of 200 µg of lectin. The maximum adsorption was observed at *dg* 87%. However, the lectin adsorption results of NANA MNWF at *dg* 40–100% were very similar. NANA MNWF with the lowest *dg* of 38% used in this experiment adsorbed 109.7 µg out of a total of 200 µg, showing an adsorption efficiency of 54.9%. The *dg* 30–100% results were similar, but it was confirmed that the adsorption amount decreased in the case of NANA MNWF with a high *dg*. These results are similar to studies that attempted to immobilize enzymes on membranes prepared through RIGP [12]. According to the results of [12], it can be confirmed that the results of the *dg* 50% membrane had higher adsorption efficiency when the enzyme was immobilized on the *dg* 50% membrane and the *dg* 100% membrane. This is thought to impede rather than induce more adsorption about the target if the density of the polymer brush is too high. Reference [12] details an experiment using a membrane with a large usable surface area to volume ratio. The membrane is a material with many advantages, but the disadvantage is that the pores are easily clogged, so it is thought that the maximum efficiency was observed at *dg* 50%. In our study, since MNWF was used, similar results were observed at a higher *dg* because the advantage of MNWF is that the pores are not easily clogged. In addition, according to [12], it was confirmed that when the GMA concentration was changed when the membranes with the same *dg* were prepared, the length of the polymer brushes of the membranes prepared at lower GMA concentrations increased. Therefore, in our research, NANA MNWF with the same *dg* but different GMA concentrations during preparation was selected, and the results were compared.

Among the samples used in the lectin adsorption experiment, adsorption efficiency was compared to the average *dg* 89% NANA MNWF (manufactured GMA concentration 4.5%) and the average *dg* 96% NANA MNWF (manufactured GMA concentration 7%). There was a 5% difference in average *dg*. However, this difference was minimal, and it was judged that the effect on the results would be negligible. This result is shown in Figure 11. Similar to the results in [12], in the case of NANA MNWF manufactured at a GMA concentration of 4.5%, the average lectin adsorption was about 10 µg higher, which is a difference in adsorption rate of about 5%. Therefore, in the case of the NANA MNWF proposed here, it can be said that the use of the MNWF produced at a concentration of 4.5% GMA with a *dg* of 40–100% showed the best efficiency.

## 4. Conclusions

In this study, NANA was introduced into MNWF by mimicking the effect of sugar chain clusters, which are densely present to increase the binding force in the process of virus adsorption to cells, and an adsorbent that successfully interacts with lectins was developed. As a functional group for introducing NANA, IDA-EDC was prepared and used. The introduction of IDA-EDC and NANA was confirmed through FT-IR peak analysis and SEM images in the MNWF of each step. In addition, to ensure the lectin adsorption performance in MNWF at each stage, the emission image of the fluorescently labeled lectin was checked, and the amount of lectin eluted with the SDS solution was compared. As a result, in the case of NANA MNWF, it was confirmed that the lectin was efficiently adsorbed. As for the maximum adsorption amount of lectin, 118.2 of 200 µg of lectin was adsorbed in *dg* 87% NANA MNWF, showing an adsorption efficiency of 59.1%. In addition, similar to the reference [12], MNWF which has an identical *dg* but has a low GMA concentration at the time of production showed a better lectin adsorption performance. This seems to be affected by the length of the polymer brush. NANA MNWF successfully adsorbed lectins in aerosol form, but this is still in the early stages of experiments. We will soon also conduct studies on the correlation between GMA concentration and lectin adsorption rate in the manufacture of GMA MNWF, the optimal coupling reaction time of functional group IDA-EDC, the exact amount of NANA introduced, and the amount of lectin adsorption per 1 g of NANA. Subsequent experiments are expected to clarify the optimal GMA concentration, IDA-EDC binding reaction conditions, and NANA introduction conditions when preparing GMA MNWF for lectin adsorption.

## Data Availability

Not applicable.

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
