# Peer review of "Functional Microfiber Nonwoven Fabric with Sialic Acid-Immobilized Polymer Brush for Capturing Lectin in Aerosol"

_polymers, 2022, doi:10.3390/polym14040663_

Round 1

Reviewer 1 Report

  1. The importance of this work needs to be highlighted in the abstract.
  2. Line 20- “A tendency to decrease….dg” - What does this line imply to a broader audience? The authors should elaborate on this in the abstract.
  3. The introduction needs more work.
  4. The section needs to be skimmed and some of the information from the materials and methods section needs to be added here (eg. selecting target lectin).
  5. A lot of terms have been introduced and their significances are not clear. For a broad audience, it will be helpful to use more generic terms or explain the significance of specific terms relevant to the paper. For example Line 65-68-“ HA (13.5 nm)……receptor [14]”
    1. The line is not clear as such and it might need to be split into two sentences.
    2. Also, please clarify the significance of the sizes of HA, influenza virus, sialic acid, Covid-19.
  • Why are ACE 2 receptors important in this case? What message are the authors trying to convey?
  1. The thought process behind selecting target lectin needs to be in the introduction section.
  2. Figure 1: Why should the broader audience care about the crystalline structure? Those two structures look quite different, and no explanation has been provided.
  3. Figure 2. The chemical structures are not uniform.
  4. The reaction method needs to be elaborated so that the reaction can be reproduced by the readers.
  5. Figure 5. How was the average IDA-EDC density calculated? The equation needs to be added in the Methods section.
  6. How were the fluorescent marker lectin images captured in Figure 6.? The description of the fluorescence microscopy method, including the microscope details, should be included in the Methods section.
  7. Figure 7. How was the lectin absorption rate measured? The details must be included in the Methods section.
  8. Figure 8 needs further explanation. Why do we see a decrease in lectin absorption with the increase in GMA concentration? Also, how was the lectin absorption rate quantified?

Author Response

Before answering, I would like to thank the reviewers for their valuable time and opinions in developing our research. We have tried to answer each question as much as possible. There is still a little more to add, but please keep an eye on the current progress.

1. The importance of this work needs to be highlighted in the abstract.

A. We added and changed the content of the front and back sections of the abstract to make the subject matter and results a little more clearly.

2. Line 20- “A tendency to decrease….dg” - What does this line imply to a broader audience? The authors should elaborate on this in the abstract.

A. We've edited this part to be more clearly.

3. The introduction needs more work.

A. To avoid misunderstandings, we have deleted all of the contents of COVID 19 and added and corrected the range of the theory and concept used in the experiment.

4.The section needs to be skimmed and some of the information from the materials and methods section needs to be added here (eg. selecting target lectin).

A. The candidates for the lectins we were going to use and the reasons for their selection were additionally described.

5. A lot of terms have been introduced and their significances are not clear. For a broad audience, it will be helpful to use more generic terms or explain the significance of specific terms relevant to the paper. For example Line 65-68-“ HA (13.5 nm)……receptor [14]”

*The line is not clear as such and it might need to be split into two sentences.

*Also, please clarify the significance of the sizes of HA, influenza virus, sialic acid, Covid-19.

A. In the case of the HA you mentioned, the full name is mentioned as Hemagglutinin in the previous sentence. In addition, since the infection process is different in the case of COVID-19, the reason mentioned in this study is only mentioned as one example of a representative epidemic. However, we have deleted the content about COVID-19 because there may be a misunderstanding that the mention of Corona 19 is also related to the adsorption of COVID-19.

*Why are ACE 2 receptors important in this case? What message are the authors trying to convey?

A. In this experiment, the ACE 2 receptor was not involved. Therefore, it has been deleted.

6. The thought process behind selecting target lectin needs to be in the introduction section.

A. We briefly expressed this content in the introduction. We also added more content in the how-to section.

7. Figure 1: Why should the broader audience care about the crystalline structure? Those two structures look quite different, and no explanation has been provided.

A. Although the crystal structures of the influenza virus and WGA are different, both have in common that they react with NANA through specific recognition. This content has been added to the [Target lectin selection] section of the text.

8. Figure 2. The chemical structures are not uniform.

A. Since the image of the middle EDC did not look uniform, it was corrected and inserted.

9. The reaction method needs to be elaborated so that the reaction can be reproduced by the readers.

A. All conditions of the experiments we performed were entered.

10. Figure 5. How was the average IDA-EDC density calculated? The equation needs to be added in the Methods section.

A. Added an equation to calculate the amount of GMA in mmol/g. Since the amount of GMA introduced and the molar conversion rate from GMA to IDA-EDC are known, the amount of IDA-EDC introduced can be confirmed.

11. How were the fluorescent marker lectin images captured in Figure 6.? The description of the fluorescence microscopy method, including the microscope details, should be included in the Methods section.

A. I deeply agree with your comments. We would also like to add this. However, since the person in charge of the device is currently absent, we promise to add this content by 01/25 or 01/26.

12. Figure 7. How was the lectin absorption rate measured? The details must be included in the Methods section.

A. We have already filled this out in Methods Section 2-2-3. 2000 ug of WGA solution 0.1ml was sprayed in the form of aerosol and adsorbed through NANA MNWF. In this case, the maximum lectin retention in 0.1 ml of WGA solution is 200 ug. Afterward, NANA MNWF was washed with deionized water to wash the adsorption residue, and lectin was extracted using 0.2 wt% of SDS solution. The lectin extracted by the SDS solution was transformed into a state that can be quantitatively measured by spectrophotometry through the BCA method. After making a calibration curve, it is quantitatively measured through a spectrophotometer.

13. Figure 8 needs further explanation. Why do we see a decrease in lectin absorption with the increase in GMA concentration? Also, how was the lectin absorption rate quantified?

A. The absorption rate of lectin was calculated as (lectin eluted from NANA MNWF/200 ug)*100[%]. The decrease in lectin absorption as the GMA concentration increased is listed in result 3.2. According to reference [12], it was reported that as the dg of MNWF increases, the density of polymer brushes increases. For GMA MNWF in our experiment, we consider that the density of the polymer brushes was too high when the dg was higher than 100%, preventing the lectins from penetrating the sialic acid-introduced portions of the polymer brushes.

Once again, we would like to thank the reviewers for their valuable time.

Reviewer 2 Report

The authors present a manuscript detailing the functionalization of nonwoven polymer fibers with sialic acid by radiation induced graft-polymerization. The manuscript is well written and readily understood. The language used is very good. Overall the presented research could be valuable to other researchers in the field of biomedical materials.  The researchers presented a good introduction of the problem and previous work by other researchers. The researchers were very careful in describing the experimental part. Therefore, these experiments could be replicated by other researchers familiar with the area. The conclusions in this manuscript are supported by the experimental results. Thus I support the publication of this manuscript.

Author Response

First of all, we would like to thank the reviewers for their valuable time and full support for the development of our research. As another reviewer pointed out, we would like to describe how we characterize the NANA MNWF we produced in a little more detail. Data from FT-IR have now been added, and information about the device that obtained the images of the previously used fluorescently labeled lectins will be added soon. Finally, we would like to thank the reviewers for their valuable time and comments.

Reviewer 3 Report

This article describes the development of a functionalized microfiber nonwoven fabric material to adsorb the influenza virus, which was prepared through a radiation-induced graft polymerization, and a NANA post-modification. The maximum lectin adsorption was observed to be 59.1% by adsorbing 118.2 μg of lectin out of a total amount of 200 μg of lectin in dg 87% NANA MNWF. This is a good initial result. It is of great significance to develop novel material for the mask in the period of COVID-19. In addition, this article has been very well written and nicely put together with a lot of care and it is also evident that the authors have attempted to be very thorough. Therefore, this work can be published in Polymers after minor revisions.

  1. In line 125, there should be cite appropriate references behind the sentence “and since there are many related kinds of literature and studies”
  2. How about the stability of IDA-EDC? From the perspective of Organic Chemistry, the IDA-EDC is not stable because the presence of secondary amines, it can occur intermolecular reactions to obtain amides.
  3. In Figure 3, How does MNWF generate radical by electron beam?
  4. In Figure 3, the structure of IDA-EDC is incorrect, hydrogen in secondary amine was lost.
  5. Can the author give an objective evaluation of the application of the materials in mask?

Author Response

Before answering, I would like to thank the reviewers for their valuable time and opinions in developing our research. We have tried to answer each question as much as possible. There is still a little more to add, but please keep an eye on the current progress.

1. In line 125, there should be cite appropriate references behind the sentence “and since there are many related kinds of literature and studies”

A. Added and corrected the appropriate reference to that document in steps 2-2-1.

2. How about the stability of IDA-EDC? From the perspective of Organic Chemistry, the IDA-EDC is not stable because the presence of secondary amines, it can occur intermolecular reactions to obtain amides.

A. To answer this question, we sought advice from Dr. Fujii, who is majoring in organic chemistry. In his opinion, IDA-EDC is unlikely to generate secondary amines and amides. Still, he suggested that it would be good to identify through the FT-IR analysis peaks accurately. Therefore, we wanted to identify the peak of the amide in FT-IR. In reference [54], we were able to identify the peaks of Amide I (1554 -1564 cm-1) and Amide II (1659-1662 cm-1). This peak also appeared in our sample. However, we think this is the peak of N-H possessed by IDA-EDC (1554-1564, 1659-1662, 3250-3400 cm-1), not that of Amide. Also, even if this is an amide, the peak is not prominent, so it is thought that the occurrence is not much. Therefore, in our opinion, IDA-EDC appears to be in a stable state.

3. In Figure 3, How does MNWF generate radical by electron beam?

A. When an electron beam is shot on a substrate, a part of the chemical structure of the substrate is in a destroyed state, and this is said to be radical. When a radical reacts with a monomer, a material with a monomer added to the substrate is prepared.

4. In Figure 3, the structure of IDA-EDC is incorrect, hydrogen in secondary amine was lost.

A. In the C2H5-N=C=N-(CH2)3-N-(CH3)2 state of EDC, some of the bonds on the left of C were bound to IDA, and the rest were expressed as they are. Please let us know if there are any parts we've missed.

5. Can the author give an objective evaluation of the application of the materials in mask?

A. As the degree of grafting increases by more than 150%, it has been confirmed that the sorbent's physical durability decreases this time, such as the external shape of the adsorbent is deformed. Therefore, if you want to manufacture it as a filter for a mask, I think it is necessary to have a method to suppress the physical deformation when each functional group is introduced. If the external frame can suppress the deformation that occurs during the introduction of functional groups and the drying process, it is judged that it can be used as an internal material such as a mask. We tried the reaction in a centrifuge tube and dried the sample on a petri dish in this experiment.  In the case of our sample, to suppress deformation, we operated to straighten the bent portions of individual samples during the washing and drying process after introducing the functional group.

Finally, we would like to thank the reviewers for their valuable time for the further development of our research.

Reviewer 4 Report

In this manuscript, the authors reported the functionalization of MNWF with sialic acid-immobilized polymer brush for the application of lectin adsorption. It was found that the degree of graft polymerization of NANA MNWF could affect the adsorption of lectin onto the fabric. The maximum adsorption amount was measured to be 59.1% with a degree grafting of 87%. It is an interesting work. This study will be helpful for the design and fabrication of functional fabric materials for biomedical applications. Based on these points, this manuscript is recommended for publication after some questions are addressed carefully.  

Special comments for the revision:

  1. It is necessary for the authors to provide more information on the novelty and significance of this work.
  2. It is necessary for the authors to add more characterization to prove the successful functionalization of MNWF at different stages. For instance, FTIR could be used.
  3. In the “experimental” part, the information on the characterization techniques should be provided.
  4. The functionalized MNWF at different stages should be checked with SEM. The current fluorescent images have very low resolution. In addition, the scale bars are not clear.
  5. In Figure 8, the authors are suggested to check the effect of more GMA concentrations on the lectin adsorption. Two sets of data are not enough for obtaining a reasonable conclusion.
  6. The part of “Conclusions” should be simplified.

Author Response

Before answering, I would like to thank the reviewers for their valuable time and opinions in developing our research. We have tried to answer each question as much as possible. There is still a little more to add, but please keep an eye on the current progress.

1. It is necessary for the authors to provide more information on the novelty and significance of this work.

A. We tried adsorption of influenza virus from readily available parent material (lectin was used as a substitute for the influenza virus in this experiment), and our method worked. We think that more experiments are still needed to find optimal conditions by controlling multiple variables. However, we also think that the proof and success of this idea alone are meaningful.

2. It is necessary for the authors to add more characterization to prove the successful functionalization of MNWF at different stages. For instance, FTIR could be used.

A. Added FT-IR data to result 3.1

3. In the “experimental” part, the information on the characterization techniques should be provided.

A. Added the phrase FT-IR analysis and the range of wavenumbers of FT-IR to the experimental part. Also, we are currently checking the conditions of the equipment used to obtain the fluorescence images. It seems that the conditions of the equipment that brought the fluorescence image can be added on 01/25 or 01/26

4. The functionalized MNWF at different stages should be checked with SEM. The current fluorescent images have very low resolution. In addition, the scale bars are not clear.

A. We edited the photo to make it more visible. And most of the previously used MWNF has been consumed, we are trying to obtain an SEM image with the newly produced MNWF. It seems that the above work can also be solved within 01/25 or 01/26.

5. In Figure 8, the authors are suggested to check the effect of more GMA concentrations on the lectin adsorption. Two sets of data are not enough for obtaining a reasonable conclusion.

A. Thoughts on this data were constructed from reference [12]. In the reference literature, although the GMA concentration used for membrane manufacturing is different, the density is low in the case of a membrane having a similar dg, however, the length of the polymer brush is increased. Referring to the above results, we have MNWFs with similar dg in our data, but it was thought that the MNWFs of the two classes had different lengths and densities of polymer brushes. And we found higher lectin adsorption in samples where the length of the polymer brush is thought to be longer, and this result was similar to the reference. Since the MNWF provided by ENEOS this time was made based only on the degree of grafting, it was not easy to control the variables. Therefore, since the part where we could suppress the variable was limited, we made data by preparing only samples prepared with different GMA concentrations under the same conditions as much as possible. In the next experiment, we plan to make more accurate data by adjusting the variables in this part and making data with more samples. We would appreciate it if you could look at the current data as an assumption stage.

6. The part of “Conclusions” should be simplified.

A. We modified the contents of "Conclusion" to be more straightforward

Finally, we would like to thank the reviewers for their valuable time and opinions for the further development of our research.

Round 2

Reviewer 1 Report

The authors have done the necessary changes. The manuscript can be accepted to this journal. 

Reviewer 4 Report

In this revised version, the authors made great improvement according to the comments and suggestions of all referees. I am satisfied with these changes and therefore recommend the publication of this manuscript at the current version.